# Liposomal Form of 2,4-Dinitrophenol Lipophilic Derivatives as a Promising Therapeutic Agent for ATP Synthesis Inhibition

**DOI:** 10.3390/nano12132162

**Published:** 2022-06-23

**Authors:** Kseniya Yu. Vlasova, Petr Ostroverkhov, Daria Vedenyapina, Tamara Yakimova, Alla Trusova, Galina Yurievna Lomakina, Stepan Sergeevich Vodopyanov, Mikhail Grin, Natalia Klyachko, Vladimir Chekhonin, Maxim Abakumov

**Affiliations:** 1Department of Medical Nanobiotechnology, Pirogov Russian National Research Medical University, 117997 Moscow, Russia; vlasova_k.y@mail.ru (K.Y.V.); chekhoninnew@yandex.ru (V.C.); 2School of Chemistry, Lomonosov Moscow State University, 119991 Moscow, Russia; lomakinagalina@yahoo.com (G.Y.L.); nlklyachko@gmail.com (N.K.); 3Department of Chemistry and Technology of Biologically Active Compounds, Medical and Organic Chemistry, Lomonosov Institute of Fine Chemical Technologies MIREA-Russian Technological University (RTU MIREA), 119571 Moscow, Russia; mrp_ost@mail.ru (P.O.); lykrecia@mail.ru (D.V.); grin@mirea.ru (M.G.); 4Faculty of Materials Science, Lomonosov Moscow State University, 119991 Moscow, Russia; yakimova_t31@inbox.ru (T.Y.); alla.trusova.fnm@gmail.com (A.T.); 5College of New Materials and Nanotechnologies, National University of Science and Technology (MISIS), 119049 Moscow, Russia; stepan.vodopianov@yandex.ru; 6V. Serbsky National Medical Research Center for Psychiatry and Narcology, 119034 Moscow, Russia

**Keywords:** obesity, non-alcoholic fatty liver disease, mitochondrial uncoupler, 2,4-dinitrophenol, liposomes, ATP synthesis inhibition

## Abstract

Mitochondrial uncoupler 2,4-dinitrophenol (2,4-DNP) is a promising antidiabetic and antiobesity agent. Its clinical use is limited by a narrow dynamic range and accumulation in non-target sensitive organs, which results in whole-body toxicity. A liposomal formulation could enable the mentioned drawbacks to be overcome and simplify the liver-targeted delivery and sustained release of 2,4-DNP. We synthesized 2,4-DNP esters with carboxylic acids of various lipophilic degrees using carboxylic acid chloride and then loaded them into liposomes. We demonstrated the effective increase in the entrapment of 2,4-DNP into liposomes when esters were used. Here, we examined the dependence of the sustained release of 2,4-DNP from liposomes on the lipid composition and LogP_oct_ of the ester. We posit that the optimal chain length of the ester should be close to the palmitic acid and the lipid membrane should be composed of phospholipids with a certain phase transition point depending on the desired release rate. The increased effect of the ATP synthesis inhibition of the liposomal forms of caproic and palmitic acid esters compared to free molecules in liver hepatocytes was demonstrated. The liposomes’ stability could well be responsible for this result. This work demonstrates promising possibilities for the liver-targeted delivery of the 2,4-DNP esters with carboxylic acids loaded into liposomes for ATP synthesis inhibition.

## 1. Introduction

Type 2 diabetes mellitus (T2DM) and non-alcoholic fatty liver disease (NAFLD) are among the most significant problems in modern medicine [1,2,3]. Independently or in combination, these chronic diseases result in high risks of disability and mortality [4]. NAFLD is increasingly becoming the most frequent liver disease affecting adults [5,6,7] and children [4]. Based on the foregoing, the discovery and development of a drug that enables the prevention and treatment of NAFLD and T2DM are extremely important [4,5,6,7]. Nowadays, a number of drugs aimed to treat T2DM are used in clinical practice, but there is no drug treatment for NAFLD [8]. According to the literature, there are no drugs approved for clinical use for the treatment of NAFLD, either as an independent disease or in combination with T2DM [9,10,11]. The NAFLD therapy is based on diet and physical exercises, as well as the selection of approved antidiabetic drugs, which may, in some cases, reduce liver inflammation. However, the effectiveness of antidiabetic drugs in the treatment of NAFLD is poorly demonstrated in clinical trials. Thus, the development of a drug for the prevention and treatment of NAFLD is relevant. One of the promising molecules that can be effective and safe in the treatment of NAFLD is 2,4-dinitrophenol (2,4-DNP), which changes the activity of mitochondria [12,13,14,15]. Indeed, 2,4-DNP acts as a carrier of a proton through the membrane of the mitochondria, scattering the gradient of mitochondrial protons and promoting the thermal dissipation of energy resulting from the oxidation of the mitochondrial substrate [16]. Previously, 2,4-DNP was widely used to lose weight, but this use was ceased due to side effects, such as high toxicity, fatal hyperthermia, etc. [17]. The severity of these side effects calls for the development of an effective liver-targeted formulation of 2,4-DNP. Currently, there are only a few experimental works focused on the development of a dosage form of 2,4-DNP for oral [18,19] and invasive administration [20]. Such formulations were effective in the treatment of NAFLD, but, unfortunately, they suffered from some limitations. For example, Shulman and co-workers synthesized and studied a liver-targeted prodrug form of 2,4-DNP for oral administration—2,4-DNP-methyl ether (DNPME) [18]. They demonstrated a promising increase in the therapeutic window for DNPME and the ratio of toxic to effective dose (50 times) [18,19]. The most significant limitations of using this prodrug are a high peak plasma concentration and rapidly released pharmacokinetics [18]. Liquid crystals in suit gel loaded with 2,4-DNP showed significant prolongation of the in vitro release [20], but the clinical use of such a formulation type is limited by its viscosity [21].

We suppose that the use of a liposomal formulation for invasive administration will allow to overcome the mentioned drawbacks and simplify the liver-targeted delivery and sustained release of 2,4-DNP. Liposomes are well-studied lipid vesicles that are broadly used in clinical applications [22,23,24,25]. Recent developments in relation to this issue allow the production of liposomes of various architectures that possess high stability, long circulation, sustained drug release, stimuli response, etc. [26,27,28]. Moreover, a number of studies have demonstrated that liposomes are likely to accumulate in the liver [29,30,31], which is essential for 2,4-DNP therapy. It is well-known that low-lipophilic molecules, such as 2,4-DNP, rapidly release from liposomes, which necessitates increasing the lipophilicity of those molecules by chemical modification [32]. In this context, we used lipophilic 2,4-DNP carboxylic acid esters due to the ease of their synthesis and well-studied metabolism [33]. Such derivatives are prodrugs, as well as DNPME, with low cytotoxicity, and they do not demonstrate uncoupler activity since the hydroxyl group is converted into an ester, which is not capable of transferring protons. Moreover, the gradual biodegradation of ester derivatives by esterases [33] leads to the prolonged release of 2,4-DNP in the cells and, as a result, to the prolonged decrease in ATP synthesis. The steric structure and composition of the derivatives prone them to incorporate into lipid membrane [34]. Since the more hydrophobic molecules interfere with the bilayer more strongly, they are also slowly released from the liposomes, which provides the molecules stability and delivery to the target. Thus, this work proposes the development of a sustained-release liposomal formulation of 2,4-DNP carboxylic acid esters.

## 2. Materials and Methods

The 2,4-Dinitrophenol (2,4-DNP), N-(3-Dimethylaminopropyl)-N′-ethylcarbodiimide (EDC), 4-(Dimethylamino)pyridine (DMAP), N,N-Dimethylformamide (DMF, ACS reagent grade), Palmitic acid, Hexanoic acid, Propionic acid were purchased from Merck Sigma-Aldrich (Saint Louis, MO, USA). Oxalyl chloride was purchased from Acros Organics (Geel, Belgium). Phosphatidyl choline from chicken egg (eggPC), 1,2-distearoyl-sn-glycero-3-phosphocholine (DSPC), 1,2-distearoyl-sn-glycero-3-phosphoethanolamine-N-methoxy(polyethylene glycol)-2000 (DSPE-PEG-OMe) were purchased from Avanti Polar Lipids (Alabaster, AL, USA). D-Luciferin sodium salt was purchased from BioVision Inc (Milpitas, CA, USA).

^1^H and ^13^C NMR spectra were registered using Bruker DPX 300 spectrometers in CDCl_3_. Signals of residual ^1^H and ^13^C nuclei were used for scale calibration. Preparative column chromatography was performed using Merck 40/60 430 silica gel sorbent.

### 2.1. Synthesis of 2,4-DNP Carboxylic Acid Esters

#### 2.1.1. Synthesis of 2,4-DNP Palmitic Acid Ester (2,4-DNP-C16) Using the Activated Ester Method

To obtain compound **2** (2,4-DNP-C16), 1-ethyl-3-(3-dimethylaminopropyl) carbodiimide (EDC) (90 mg, 0.58 mmol) and 4-dimethylaminopyridine (DMAP) (19.5 mg, 0.16 mmol) were added to a solution of palmitic acid (100 mg, 0.39 mmol) in chloroform (5 mL). The mixture was cooled to 5 °C and stirred over 30 min. Then, an aliquot of 2,4-DNP (72 mg, 0.39 mmol) was added to the reaction mixture. The reaction was continued with constant stirring while cooling for 6 h. The reaction progress was monitored by thin-layer chromatography (TLC) (see Appendix A). The organic solvent was removed by reduced-pressure evaporation under vacuum. The final product (compound **2**) was isolated in two steps by washing with cold water followed by column chromatography. Recrystallization from hexane provided 34 mg (yield 34%) of a pale-yellow crystal product. ^1^H NMR (300 MHz, CDCl_3_, δ, ppm): 8.99 (H, d, *J* = 2.7 Hz, 3′-CH), 8.39 (H, dd, *J* = 9.3, 2.7 Hz, 5′-CH), 7.30–7.24 (H, m, 6′-CH), 2.27 (2H, t, *J* = 7.5 Hz, 2-CH), 1.55 (2H, m, 3-CH), 1.18 (23H, m, 4-CH2–15-CH2) 0.80 (3H, t, *J* = 6.7 Hz, 16-CH3).

#### 2.1.2. Synthesis of 2,4-DNP-C16 Ester Using Palmitic Acid Chloride

Palmitic acid chloride was obtained by the reaction of oxalyl chloride (50 μL, 0.58 mmol) with palmitic acid (100 mg, 0.39 mmol) dissolved in chloroform (5 mL) in the presence of catalytic amount of dimethylformamide (DMF) (2 μL, 0.012). The mixture was cooled to 5 °C and stirred over 30 min. After the reaction was completed, the solvents were removed under reduced pressure using a rotary evaporator.

Next, the obtained palmitic acid chloride (melting point 12 °C) was redissolved in chloroform. A solution of 2,4-DNP (72 mg, 0.39 mmol) in chloroform (2 mL) was mixed with triethylamine (142 μL, 0.39 mmol), providing a bright-yellow solution due to the formation of phenolate. Then, the produced solution was added dropwise to the palmitic acid chloride obtained earlier. The resulting mixture was boiled at 50–55 °C under vigorous stirring. The progress of the reaction was monitored by TLC (see Appendix A). The organic solvent was removed by reduced-pressure evaporation under vacuum. The final product (compound **2**) was isolated in two steps by washing with cold water followed by column chromatography. Recrystallization from hexane provided 73 mg (yield 73%) of a pale-yellow crystal product. ^1^H NMR (300 MHz, CDCl_3_, δ, ppm): 8.95 (H, d, *J* = 2.1 Hz, 3′-CH), 8.51 (H, dd, *J* = 8.8, 2.5 Hz, 5′-CH), 7.46 (H d, *J* = 8.9 Hz, 6′-CH), 2.68 (2H t, *J* = 7.5 Hz, 2-CH2), 1.86–1.70 (2H m, 3-CH2), 1.45–1.20 (23H, m, 4-CH2–15-CH2), 0.88 (3H, t, *J* = 5.5 Hz, 6-CH3). ^13^C NMR (300 MHz, CDCl_3_, δ, ppm): 170.58, 148.90, 145.17, 141.96, 129.04, 126.80, 121.83, 34.16, 32.08, 29.83 (5C), 29.73, 29.56, 29.51, 29.34, 29.13, 24.48, 22.84, 14.25.

#### 2.1.3. Synthesis of 2,4-Dinitrophenol Caproic Acid Ester (2,4-DNP-C6)

2,4-DNP-C6 (compound **3**) was synthesized using acid chloride method as described above for compound **2**. Amounts of reagents are presented in Table 1.

The progress of the reaction was monitored by TLC (see Appendix A). Recrystallization from hexane provided 495.6 mg (yield 70.5%) of a pale-yellow product.

^1^H NMR (300 MHz, CDCl_3_, δ, ppm): 8.94 (H, d, *J* = 2.3 Hz, 3′-CH), 8.51 (H, dd, *J* = 8.9, 5′-CH), 7.47 (H d, *J* = 9 Hz, 6′-CH), 2.68 (2H t, *J* = 7.5 Hz, 2-CH2), 1.79 (2H p, *J* = 7.7 Hz, 3-CH2), 1.49–1.32 (4H, m, 4-CH2, 5-CH2), 0.94 (3H, t, *J* = 6.5 Hz, 6-CH3). ^13^C NMR (300 MHz, CDCl_3_, δ, ppm): 170.56, 148.87, 145.16, 141.95, 129.05, 126.79, 121.81, 34.10, 31.24, 24.13, 22.38, 13.97. EI-MS (m/z): 71.1 (48.9%), 99.1 (100%), 154.0 (7.43%), 184.0 (2.92%), 284 (M+, <1%).

#### 2.1.4. Synthesis of 2,4-Dinitrophenol Propionic Acid Ester (2,4-DNP-C3)

2,4-DNP-C3 (compound **4**) was synthesized using acid chloride method as described above for compound **2**. Amounts of reagents are presented in Table 2.

The progress of the reaction was monitored by TLC (see Appendix A). Recrystallization from hexane provided 200.5 mg (yield 78%) of a pale-yellow product.

^1^H NMR (300 MHz, CDCl_3_, δ, ppm): 8.96 (H, d, *J* = 2.4 Hz, 3′-CH), 8.55–8.49 (H, m, 5′-CH), 7.48 (H, d, *J* = 8.9 Hz, 6′-CH), 2.73 (2H, q, *J* = 7.5 Hz, 2-CH2), 1.30 (3H, t, *J* = 7.5 Hz, 3-CH3). ^13^C NMR (300 MHz, CDCl_3_, δ, ppm): 171.29, 148.91, 145.16, 141.85, 129.14, 126.80, 121.89, 27.70, 8.70. EI-MS (m/z): 57.1 (100%), 154.0 (5.36%), 184.0 (1.2%), 240 (M+, <1%).

### 2.2. Gas Chromatography–Mass Spectrometry (GC–MS)

GC–MS analyses were registered using Agilent 7820a (GC) and Agilent 5977b (MS) devices. Column: HP-5ms 30 m × 0.25 mm × 0.25 μm, carrier gas: helium, permanent flow: 1 mL/min. A temperature-programmed GC analysis was performed starting at 250 °C, with a holding time of 1 min, and then increased to 310 °C, at 30 °C/min, with a final hold time of 2 min.

### 2.3. Octanol–Water Partition Coefficients Measurements (LogP)

Lipophilicity of the obtained 2,4-DNP derivatives was measured by octanol–water equilibrium method [35]. Before the experiment, we mixed octanol and water in a flask, shook it vigorously, and then waited for the two phases separation. After reaching of equilibrium, the two phases are mutually saturated with one another. Next, 0.5 mg of each test compound was dissolved in 1 mL of octanol phase, and then 1 mL of distillated water phase was added (pH 5.5) and properly stirred for 1 min three times. The mixture was incubated under vigorous stirring for 1 h. Next, the mixture was centrifugated to separate octanol–water phases. The concentration of the compound was determined in octanol and water using HPLC method. The octanol–water partition coefficient was estimated as follows:(1)$$LogP=Log\frac{\left[C\right]octanol}{\left[C\right]water}$$

### 2.4. 2,4-DNP Concentration Measurement

Concentration of 2,4-DNP and its esters was measured using high-performance liquid chromatography (HPLC) (Knauer Smart line, Berlin, Germany) with UV-detector and a reversed-phase shim-pack 100-5-C18 column (Kromasil, 250 nm, 4.6 nm, particle size 5 μm). The mobile phase was a mixture of acetonitrile and 0.1% citric acid water solution (60/40, *v*/*v*). The mobile phase flow-rate was 0.8 mL/min, the injection volume was 20 mL, and the detection wavelength was set at 275 nm. Chromatograms were analyzed using ChromGate version 3.3.2 software (Berlin, Germany). The retention time is 4.5 min for 2,4-DNP and 6.5 min for esters.

### 2.5. Liposomal Formulation

Liposomes were prepared using the standard rehydration thin film method. In brief, appropriate amounts of lipids and, in some cases, 2,4-DNP derivative, were placed in a round-bottom flask and dissolved in chloroform. Then, the solvent was removed under vacuum to produce a thin lipid film. Formed film was rehydrated with 10 mM sodium phosphate buffer (PBS, pH 7.4) or, in some cases, with 2,4-DNP solution in 10 mM PBS (1 mg/mL) and extruded through carbon membranes with 0.4, 0.2 µm pore size at room temperature (for eggPC liposomes) or at 65 °C (for DSPC liposomes). The unloaded active molecules were removed by gel-filtration using illustra^TM^ NAP-25 column (Sephadex G-25 DNA Grade, GE Healthcare, Little Chalfont, UK). 2,4-DNP and its derivatives content in final solution were quantified by HPLC. The lipids content was measured spectrophotometrically as described in Ref. [36]. The hydrodynamic diameter was determined by dynamic light scattering (DLS) using a Malvern Zetasizer Nano ZS (Malvern Instruments, Westborough, MA, USA) in 10 mM PBS (pH 7.4). The loading capacity was estimated as follows:(2)$$LC=\frac{\left[2,4-DNP\right],\;\mathrm{mol}}{\left[2,4-DNP\right],\mathrm{mol}+\left[lipids\right],\;\mathrm{mol}}\times 100\%$$

### 2.6. Stability Studies of Liposome

Active molecules release from liposomes was studied in 10 mM PBS (pH 7.4) at 37 °C using Spectra/Por Float-A-Lyzer G2 dialysis systems (MWCO 50 kDa) and expressed as percent of total vs. time. Briefly, 0.5 mL of the sample was added to dialysis tube and the assembly was placed in a 15 mL of 10 mM PBS (pH 7.4) and stirred for 24 h. At specific time points, 1 mL aliquots were removed from the medium outside the dialysis membrane and replaced with same volume of the fresh buffer. The amount of free 2,4-DNP was measured by HPLC method. Hydrodynamic diameter stability was studied in 10 mM PBS (pH 7.4) at 37 °C using a Malvern Zetasizer Nano ZS (Malvern Instruments, Westborough, MA, USA).

### 2.7. In Vitro Test

#### 2.7.1. Cells

Murine breast cancer cell line expressing luciferase (4T1-Luc) and hepatocytes isolated from the normal liver of a 3-month-old mouse (AML12) were purchased from ATCC, (CRL-2539-LUC2 and CRL-2254). The cells were cultivated in RPMI 1640 or DMEM-F12 with 10% FBS, 4.5 g/L glucose, 100 U/mL penicillin, 100 µg/mL streptomycin and 0.25 µg/mL Gibco amphotericin at 37 °C, 5% CO_2_.

#### 2.7.2. Cytotoxicity Test

4T1-Luc or AML12 cells were seeded in a 96-well plate (1 × 10^4^ cells/well) in 200 µL/well of complete RPMI or DMEM-F12 medium and incubated at 37 °C with 5% CO_2_ for 48 h. Then, the medium was replaced with 100 µL/well RPMI or DMEM-F12 medium in the absence (control) or presence of various amounts of the tested compounds and incubated at 37 °C with 5% CO_2_ for 24 h; 10 µL/well WST-1 solution (CELLPRO-RO Roche, St. Louis, MO, USA) was added to each well and incubated under culture conditions for 2 h. The absorbance of the samples was measured at 450 nm.

#### 2.7.3. Mitopotential Detection

Mitopotential changes in cells were detected using the Muse MitoPotential Assay (Merck Millipore; Darmstadt, Germany). 4T1-Luc cells were harvested for 48 h before the measurements, and then cells were dissociated with TrypLE™ Express (Gibco, Grand Island, NY, USA) and suspended in the complete RPMI medium. 4T1-Luc cells were treated with DMSO (1%, control sample), 2,4-DNP (compound **1**) (200 μM), or 2,4-DNP-C16 ester (compound **2**) (200 μM), or 2,4-DNP-C6 ester (compound **3**) (200 μM), or 2,4-DNP-C3 ester (compound **4**) (200 μM) for 5 min or 90 min. According to the manufacturer’s instructions, the MitoPotential working solution was added, and the cell suspensions were incubated at 37 °C for 20 min, followed by the addition of the Muse MitoPotential 7-AAD dye and incubation at room temperature for 5 min. The mitopotential of the samples was assessed by flow cytometry (Muse Cell Analyzer, Merck Millipore; Darmstadt, Germany). The percentages of depolarized living cells were analyzed relative to the control.

#### 2.7.4. Luciferin–Luciferase Coupled Assay

ATP synthesis in real time in vitro was monitored by modified luciferin–luciferase assay [37,38] using 4T1-Luc cells expressing luciferase. Briefly, 4T1-Luc cells were seeded in black 96-well plates (Greiner, France) (1 × 10^4^ cells/well) 48 h before the experiment. Cell medium was replaced with the mixtures (150 µL/well) of D-luciferin (150 µg/mL, saturating concentration) alone (control) and with examined substance (200 µM of free 2,4-DNP or its derivatives or their liposomal forms) in RPMI 1640 medium (10% FBS, 4.5 g/L glucose, 100 U/mL penicillin, 100 µg/mL streptomycin и 0.25 µg/mL Gibco amphotericin), plate was put into the Luminometer chamber (EnSpire multimode plate reader, Perkin Elmer, USA), and kinetic of luciferin–luciferase reaction was detected at 37 °C (flashes/time 0.1 s). 2,4-DNP esters were dissolved in DMSO before the use. The total concentration of DMSO in final solution per well was 1%.

The inhibition effect was expressed as
(3)$$Decrease\;of\;luminescence=\frac{{\left[RLU\right]}_{max}-{\left[RLU\right]}_{t}}{{\left[RLU\right]}_{max}}\times 100\%,$$
where [*RLU*]*_max_*—relative luminescence units at the maximum of the kinetic profile; *[RLU]_t_*—relative luminescence units at certain time point.

#### 2.7.5. Measurement of Total ATP

The ATP levels were determined by a bioluminescence assay that measures the signal formed by the oxidation of luciferin, catalyzed by luciferase in the presence of ATP. 4T1-Luc or AML12 cells were seeded into 96-well plate 48 h before the experiment. Cell medium was replaced with the mixtures of medium alone (control) and with examined substances or their liposomal forms (200 µM of free 2,4-DNP or its derivatives) in RPMI 1640 or DMEM-F12 medium (10% FBS, 4.5 g/L glucose, 100 U/mL penicillin, 100 µg/mL streptomycin и 0.25 µg/mL Gibco amphotericin). Free molecules were dissolved in DMSO. The final concentration of DMSO was 1% per well. After 2 h or 4 h incubation, the total ATP content in the medium and cells was determined as follows: a 20 μL sample of the cell suspension was taken into a test tube, and 180 μL of DMSO was added. After 1 min incubation, 20 μL of the obtained extract was taken into a polystyrene microcuvette (cat. N507050, Grenier, France), 100 μL of ATP-reagent (the mixture of luciferase, D-luciferin, MgSO_4_, and buffer) was added and bioluminescence signal was detected (for 30 s) using luminometer FB-12 (Berthold Detection Systems GmbH, Pforzheim, Germany). The mean value of the signal *I_extract_* was calculated. The bioluminescence signal was measured in a similar manner in the ATP control solution (*I_control_*). The ATP concentration (*ATP_tot_*) was calculated using the formula:(4)$$\left[ATP_{tot}\right]=10\times \left[ATP_{control}\right]\frac{I_{extract}}{I_{control}},$$
where [*ATP_control_*] = 3.75 nM in 90% DMSO, coefficient 10 is a dilution coefficient.

The results were expressed as
(5)$$Decrease\;of\;ATP\;concentration=\frac{{\left[ATP\right]}_{control}-\left[ATP\right]}{{\left[ATP\right]}_{control}}\times 100\%,$$
where [*ATP_control_*]—ATP content in the untreated cells (control); [*ATP*]—ATP content at certain time point in the treated cells.

### 2.8. Statistics

Statistical comparison studies were carried out using Student’s *t*-test. All tests were performed using Origin version 9.1.0 (64-bit) (Northampton, MA, USA).

## 3. Results

### 3.1. Synthesis of 2,4-DNP Esters

We have chosen the 2,4-DNP carboxylic acid esters derivatives for loading into the liposomal formulation based on the literature [32,33], as well as their improved lipophilicity in comparison to 2,4-DNP. A large number of methods for producing esters based on phenol [39,40,41] have been described, and the most common approaches are the method including the stage of obtaining the acid chloride as the acylating agent [42,43] and the method of activated esters [40,42,44] due to their easy implementation and high efficiency. However, the 2,4-DNP molecule has steric hindrances for the esterification reaction. In addition, the presence of an electron-acceptor group near the hydroxyl group of phenol reduces its reactivity in nucleophilic substitution reactions.

Therefore, a comparison of the efficiency of various methods for the preparation of 2,4-DNP esters is one of the objectives of this work. The determination of the most effective synthesis method was carried out using the 2,4-DNP palmitic ester since the intermediate products formed in the reaction of the nucleophilic substitution with carboxylic acid are the most stable in the case of a large alkyl substituent.

We found that the yield of the target product was more than two times higher when palmitic acid chloride was used in comparison to the reaction through an activated ester (Figure 1).

Moreover, we faced some difficulties in purifying the target product. We have chosen the two-step system of purification and isolation of 2,4-DNP esters. The first step was a filtration through a Schott filter under reduced pressure with the use of cold water as an eluent. All the esters were preliminary dissolved in cold water and kept in freezer for 1 h for better separation of the esters from the free 2,4-DNP. The crystalline precipitate was washed off with chloroform.

The second step was the column chromatography. This method was chosen due to the large loading capacity, the relative simplicity, and the routine nature of the method itself. The most significant problem we encountered during the purification of the final products from the unreacted reagents and intermediate products was the similarity of these substances (especially phenol and its derivatives) regarding many physicochemical parameters, such as solubility, color, melting and boiling points, and, as a consequence, chromatographic mobility. A number of effective systems, according to literary sources [45], such as benzene/acetic acid = 80/20 (*v*/*v*), pure benzene, and pure acetonitrile, did not show efficiency in the separation of 2,4-DNP esters and free 2,4-DNP (see Appendix A).

Based on the literature [46] and our experimental data, the following most suitable eluent system was identified: hexane/chloroform = 3/2 (*v*/*v*). Thus, the reaction mixture obtained after removing the solvents was recrystallized and dissolved in a minimum volume of hexane and loaded onto the column. During the elution, all the fractions with the same composition were collected and combined. To collect the 2,4-DNP derivatives from the reaction mixture, a stepwise gradient increase in the polarity of the eluent was used. The separation resulted in two different fractions. Unfortunately, the target product of the reaction was contained in both of the fractions, and the second fraction was a mixture of ester with free 2,4-DNP (see Appendix A). To extract the target ester from the second fraction, the separation process was repeated under the same conditions. After the purification, the qualitatively separated 2,4-DNP ester and free 2,4-DNP fractions were obtained.

The structure of compound **2** was studied by NMR spectroscopy. The ^1^H NMR spectrum showed two different sets of signals corresponding to the protons in the alkyl substituent in area 0.5–2.5 ppm and the signals assigned to the benzene ring in area 7–9 ppm (see Appendix A). It should be mentioned that the signal corresponding to the protons in the hydroxyl group is absent on the 2,4-DNP palmitic ester spectrum, unlike the dinitrophenol spectrum (see Appendix A).

The synthesis of an ester of caproic acid and 2,4-DNP was carried out using the caproic acid chloride method due to its greater efficiency, as shown above (Figure 1b). The structure of compound **3** was studied by NMR spectroscopy. The ^1^H NMR spectrum showed two different sets of signals corresponding to the protons in the alkyl substituent in area 0.5–3 ppm and the signals assigned to the benzene ring in area 7–9 ppm (see Appendix A).

The synthesis of an ester of propionic acid and 2,4-DNP was carried out using the propionic acid chloride method (Figure 1b). The structure of compound **4** was studied by NMR spectroscopy. The ^1^H NMR spectrum showed two different sets of signals corresponding to the protons in the alkyl substituent in area 1–3 ppm and the signals assigned to the benzene ring in area 7–9 ppm (see Appendix A).

In addition to the ^1^H NMR spectroscopy, ^13^C NMR spectroscopy was used to confirm the structure of compounds **2**, **3**, and **4** (see Appendix A). According to the spectra of all three compounds, the number of signals coincides with the number of carbon atoms in the corresponding compound. It is also possible to interpret most of the signals using literature data without additional research. The spectra showed a group of signals from carbons in the benzene ring (in the range of 120–150 ppm) and in the composition of the carboxylic acid fragment (in the range of 5–35 ppm and 170 ppm). Moreover, the GC–MS analysis demonstrated the presence of ionized molecules and characteristic fragments of the studied compounds, which were formed during the ionization process signals on the spectra of compounds **3** and **4** (see Appendix A).

Octanol–water partition coefficients of the synthesized derivatives were tested. These tests revealed the increase in lipophilicity in a row of substitutes C3-C6-C16 (Table 3) that are in line with the theoretically estimated LogP_oct_ (see Appendix A).

### 3.2. In Vitro Tests of 2,4-DNP Esters

The cytotoxicity of the synthesized derivatives on the 4T1-Luc and AML12 cell lines was determined using WST-1 test (Figure 1). The 24 h incubation IC_50_ of the 2,4-DNP-C3, 2,4-DNP-C6, and 2,4-DNP-C16 esters was lower (~100–150 µM on 4T1-Luc and ~200 µM on AML12) than one of the 2,4-DNP (~400 µM on 4T1-Luc and ~500–600 µM on AML12) (Figure 1). The obtained IC_50_ value of the 2,4-DNP is similar to the literature-estimated IC_50_ value reported in Refs. [47,48]. For further study of the compounds on uncoupler activity, we used them at the concentration of 200 µM, which allowed us to detect the effect at a short incubation time and exclude the parallel metabolic processes influence.

For a qualitative analysis of the ability of the produced derivatives to inhibit the oxidative phosphorylation, we used a method based on a luciferin–luciferase coupled assay [37,38]. According to the method, we followed the luminescence signal produced during the reaction of luciferase expressed by the 4T1-Luc cells themselves and D-luciferin as a substrate. The enzymatic reaction depends on the ATP content. The 2,4-DNP as a protonophore eliminates the mitochondrial proton gradient and inhibits the phosphorylation, resulting in the reduction in the ATP production [13]. Thus, we should observe a decrease in the luminescence level at the luciferin–luciferase reaction after the 2,4-DNP addition due to the decrease in ATP production. The kinetic curves profiles of the tested molecules are presented in Figure 2A.

D-luciferin is well-known to diffuse through the cell membrane [38,49]. We observed a rapid increase in the luminescence signal (during 30 min, Figure 2A) after the addition of the D-luciferin mixture with 2,4-DNP or its derivatives compared to the control. We associated this effect with the increase in the cells’ membrane permeability [50,51] and, as a result, the increase in the entrance of luciferin into the cells and initiation of the luciferin–luciferase reaction. Notably, the luminescence signal of the untreated with uncoupler cells (control) increases until around 20 min, at which point it plateaus, with small fluctuations until 240 min. The addition of the 2,4-DNP and its esters to the cells with luciferin on the plateau affected the luminescence signal level compared to the untreated control, which proved the influence of the tested molecules on the D-luciferin diffusion through the cell membrane (see Appendix A). Moreover, the comparison of the luminescence signal in the cells and in the external medium above the cells (see Appendix A) showed that the reaction took place within the cell and the release of luciferase from the cells was not observed. Meanwhile, the addition of a fluorescent dead cell marker 7-ADD (MitoPotential assay, the Muse) to the treated with the studied compounds cells also did not indicate significant changes in the membrane structural integrity compared to the untreated control upon 30 min and 120 min incubation (Figure 3A). Moreover, the mitochondrial potential studies revealed that, after the cells treatment, our compounds at the studied concentrations caused a slight cell depolarization process (Figure 3B, MitoPotential assay, the Muse). These results are in good agreement with the literature [52,53]. Behnel, H. J. et al., (1980) [52] described a biphasic process of membrane depolarization by 2,4-DNP: a short lag phase (i) is followed by a depolarization, reaching a minimum potential after 15 min. The second phase (ii) leads to a new equilibrium potential by a partial recovery or to a further depolarization to near-zero potential. The first rapid depolarization may be overcome partially after 15 min at concentrations up to 200 µM 2,4-DNP. Thus, as the decrease in the mitopotential is an indicator of mitochondrial dysfunction and a hallmark for apoptosis, we can conclude that the tested compounds did not destroy the mitochondria or cells during the study.

We assumed that the maximum on the kinetic curve matched the highest concentrations of D-luciferin and ATP in the reaction. The following luminescence signal decrease indicated the consumption of the initial ATP for luciferin oxidation at the parallel process of the inhibition of the oxidative phosphorylation by 2,4-DNP. To compare the effect of each sample, we estimated the percent of the luminescence decrease during the certain time frames (2 h and 4 h) after the addition of the examined sample from the maximum (Figure 2). There was a significant difference in the ATP synthesis inhibition among the 2,4-DNP derivatives (Figure 2A). Surprisingly, the 2,4-DNP-C3 (compound **4**) and 2,4-DNP-C6 (compound **3**) esters showed higher inhibition effects in comparison to the free 2,4-DNP. The 2,4-DNP-C6 (compound **3**) showed sustained inhibition of ATP synthesis (Figure 2A). The 2,4-DNP-C16 (compound **2**) showed a neglectable decrease in the ATP level, which could be caused by the two involved processes: (1) the slow hydrolysis of the ester and, as a result, the low free 2,4-DNP concentration, and (2) the fatty acid oxidation of the freed palmitic acid that increases the ATP production. However, the experiments with free palmitic, caproic, and propionic acids did not show an increase in the ATP level (see Appendix A) during the reaction. Thus, we are of the opinion that the hydrolysis mainly could level the action of the 2,4-DNP esters. To prove that the decrease in the luminescence in the kinetic profiles refers to the reduction in the ATP content, we conducted the experiment when the cells’ incubation (2 h or 4 h) with the uncouplers was followed by the luciferin–luciferase coupled reaction initiation by the addition of D-luciferin [36] (see Appendix A). According to the results, this approach showed the relevant or similar effect as was estimated from the kinetic profile (Figure 2B). Moreover, the control experiment showed the addition of the 2,4-DNP at the concentration of 200 µM to the luciferin–luciferase mixture did not influence the luciferase activity (data are not presented). The direct estimation of the ATP content in the cells revealed the same tendency as was described above (see Appendix A).

### 3.3. Liposomal Formulations of 2,4-DNP Derivatives

Next, we loaded the synthesized molecules into the liposomes based on lipids with different melting points. As many studies showed, the liposomes’ loading capacity, stability, time circulation, and ability of sustained release depend on the lipid membrane phase state [22]. We decided to compare the DSPC (melting point 54–56 °C [54]) and eggPC (melting point −15–+4 °C [54]) liposomes grafted with DSPE-PEG-OMe, which were in gel and liquid crystalline phase states at 37 °C, respectively. The composition and loading capacity (LC) of the samples are presented in Table 4 (see Appendix A).

All the samples were prepared via the hydration of the lipid thin film, followed by extrusion. In the case of the DSPC liposomes, the suspensions were heated up to 65 °C to obtain homogeneous dispersion. The initial molar ratio of esters to lipids was chosen as 1 to 6, respectively, to exclude a significant effect on the membrane structure (for example, permeability) and micelles formation [34].

According to the data from Table 4, the LC of the produced liposomes was significantly higher for the 2,4-DNP ester in comparison to the free 2,4-DNP. Further comparison of the LC of the liposomes of different lipid compositions showed that the esters were more prone to incorporate into the eggPC bilayer, especially in the case of the 2,4-DNP-C3 (compound **4**). An examination of the ester’s stability in 10 mM PBS (pH 7.4) at room temperature and at 65 °C revealed accelerated esters hydrolysis at a high temperature (see Appendix A). Taking in account the LC data (Table 4), the most unstable ester was 2,4-DNP-C3 (compound **4**). Thus, during the incorporation into the DSPC liposomes, where the heating step is unavoidable, we are faced with a higher rate of hydrolysis, which does not allow obtaining a stable liposomal formulation in the case of 2,4-DNP-C3.

Stability studies using dialysis systems (MWCO 50 kDa) revealed that 60–70% of what was loaded into eggPC liposomes 2,4-DNP and 2,4-DNP-C3, 2,4-DNP-C6 was released during the first 2 h at 37 °C (Figure 4A). The analysis did not confirm any significant differences in the release rates between the eggPC (liquid-crystalline phase state) and DSPC (gel phase state) liposomal formulations of 2,4-DNP and 2,4-DNP-C6 (Figure 4A,B).

The release of 2,4-DNP-C16 from eggPC liposomes was rather slow (40–45% during 2 h at 37 °C) in comparison to the other esters (Figure 4A). Remarkably, the release profile significantly changed for the 2,4-DNP-C16 incorporated in the DSPC liposomes (Figure 4B). We observed a pronounced sustained release of 2,4-DNP-C16 from the DSPC liposomes (at least during 4 h). It was caused by the gel phase state of the bilayer at 37 °C and, as a result, the higher membrane density and the decreased lateral diffusion of the lipids [55], which reduced the leakage of the large lipophilic molecules.

The hydrodynamic diameters of the studied samples did not change upon incubation in 10 mM PBS at 37 °C, at least during 24 h (Figure 4C,D). Moreover, the sizes of all the samples were stable upon storage in 10 mM PBS at 4 °C during a month.

### 3.4. In Vitro Test of Liposomal Formulations

In vitro studies on 4T1-Luc cells showed changes in the kinetic profiles of the tested molecules after their encapsulation into liposomes (Figure 5, Appendix A).

A comparison of the kinetic curves of the free molecules (Figure 2A) and molecules loaded into eggPC liposomes (Appendix A) showed the latter reaching the plateau for the last ones, which indicated sustained release of the active molecules from liposomes. According to the degree of luminescence decrease, the most effective formulations were 2,4-DNP (compound **1**) and 2,4-DNP-C3 (compound **4**) loaded into eggPC liposomes (Figure 5). The analysis confirmed differences in the in vitro kinetic profiles between the eggPC and DSPC liposomal formulations of 2,4-DNP (compound **1**), 2,4-DNP-C6 (compound **3**), 2,4-DNP-C16 (compound **2**) (Figure 5, Appendix A). We observed reduced inhibition of the ATP synthesis effect of molecules loaded into DSPC liposomes (Figure 5). This is in good agreement with the release profiles (Figure 5A,B). Notably, the eggPC liposomal forms of 2,4-DNP and 2,4-DNP-C16 showed the increased uncoupler activity after 4 h incubation compared to their free forms. Meanwhile, the addition of empty liposomes of the same composition to the cells led to an ATP level increase in the cells (see Appendix A). According to the cytotoxicity test (see Appendix A), the encapsulation of compound **1** and compound **2** led to an increase in toxicity, probably due to the increase in the bioavailability of the tested molecules.

The treatment of healthy AML12 cells with the produced liposomes also led to a decrease in the ATP level (Figure 6). The maximal inhibition of the ATP synthesis effect was observed in the case of 2,4-DNP-C6 and 2,4-DNP-C16 loaded into eggPC liposomes. The analysis of the free derivatives at the same concentration (200 µM) did not reveal a reduction in the ATP concentration in the AML12 cells (see Appendix A). Meanwhile, the control experiments with unloaded liposomes showed that empty vesicles did not increase the ATP level after 2 h of incubation and did increase the ATP level after the 4 h point (see Appendix A). Thus, the inhibition effect by the liposomal form of molecules might be explained by the increase in the bioavailability of the tested molecules, as demonstrated on the 4T1-Luc cells. Moreover, further tests revealed that 2,4-DNP-C6 and 2,4-DNP-C16 loaded into DSPC liposomes showed a lower ATP synthesis inhibition effect compared to the eggPC liposomes. These data are in line with our data on 4T1-Luc cells (Figure 5) and could be explained as a reduced rate of active molecules release from the liposomes in the cells.

## 4. Discussion

In recent years, there has been growing interest in mitochondrial uncouplers as promising antidiabetic and antiobesity agents [16]. According to the numerous studies of 2,4-DNP as a therapeutic molecule, a sufficiently safer derivative should be developed with a wider dynamic range and reduced accumulation in non-target sensitive organs [13]. Shulman and coworkers reported on an effective prodrug-form 2,4-DNP-methyl ether (DNPME) [18]. The authors indicated some limitations regarding the oral administration of that form. It was quite difficult to carry out the controlled delivery and sustained release of the drug in the target organ [19]. In this context, we tried to produce liposomes, known as sustained-release and accumulated-in-liver drug delivery systems [23,24], loaded with lipophilic 2,4-DNP derivatives.

First, we focused our work on the synthesis of lipophilic 2,4-DNP carboxylic acid esters due to their predictable mechanism of action and steric structure and composition, which make the esters prone to incorporating into the lipid membrane [33,34]. Thus, we provided the optimized method of the synthesis of 2,4-DNP acid esters. Our procedure is high-yield (approximately 70% and over) and rather universal for the production of 2,4-DNP esters with carboxylic acids of various chain lengths.

In vitro tests of the synthesized esters using 4T1-Luc cells demonstrated the correlation between the level of ATP synthesis inhibition for a certain time frame (4 h) and the chain length of the substitute. The shorter the carboxylic acid chain that was used, the more pronounced the inhibition effect was. It was also important that free carboxylic acids did not affect the reaction kinetics. These results could refer to different rates of esters metabolism in the cells by esterases [33]. Our findings would simplify the design of a 2,4-DNP acid ester with the desired sustained release of protonophore for future studies. It should be noted that the tested derivatives do not lead to the dramatic cell depolarization at the concentration of the significant ATP synthesis inhibition. Moreover, it was shown previously that, although chronic, severe uncoupling of oxidative phosphorylation is typically deleterious to normal mitochondrial and cell function; recent evidence suggests that mild uncoupling might serve a protective role [56,57].

Another task was to produce liposomes loaded with the synthesized esters. The loading capacity data showed an increase in encapsulation of approximately three times for 2,4-DNP esters with caproic and palmitic acids in comparison to free 2,4-DNP. This result has demonstrated the logic of the general aim of our work to improve the loading of 2,4-DNP. Unfortunately, the loading efficacy depended also on the lipid composition, particularly on the preparation conditions. Thus, the increase in the temperature affected the esters’ stability and accelerated their hydrolysis. In this case, we also observed the evident correlation between the hydrolysis stage and the chain length of the substitute. The most unstable ester among the studied derivatives in this work was the 2,4-DNP propionic acid ester, which resulted in the low loading of protonophore into DSPC liposomes.

The sustained release of 2,4-DNP from eggPC liposomes was observed for all the samples. A more significant sustained-release effect was observed in the case of liposomes loaded with ester with palmitic acid (40–45% for 2 h at 37 °C). Surprisingly, the rates of protonophore release from eggPC liposomes loaded with esters with propionic or caproic acids and free 2,4-DNP were similar (60–70% for 2 h at 37 °C). Notably, in vitro tests using 4T1-Luc and AML12 cells showed that the effect of liposomes loaded with derivatives depends on the chain length of the substitute and type of cells. Thus, the inhibition effect of the liposomal forms of 2,4-DNP-C3 and 2,4-DNP-C6 was higher and more pronounced in comparison to the liposomal form of 2,4-DNP-C16 ester in the case of murine breast cancer 4T1-Luc cells. Moreover, the inhibition effect of the liposomal form of 2,4-DNP-C16 was higher in comparison to the free 2,4-DNP-C16 ester. This could occur due to the increased bioavailability of the 2,4-DNP-C16 entrapped in the liposomes, which was demonstrated by the increased cytotoxicity of liposomal 2,4-DNP-C16 (see Appendix A). In the case of hepatocytes isolated from the normal liver of a 3-month-old mouse AML12, the most pronounced effect was observed when the cells were treated with liposomal forms of 2,4-DNP-C6 and 2,4-DNP-C16. In contrast to these results, the treatment of AML12 cells with free active molecules at the same concentration as in the liposomal forms (200 µM) did not lead to a reduction in the ATP level (see Appendix A). In our opinion, the higher level of the 2,4-DNP-C6 and 2,4-DNP-C16 liposomal form effect, compared to 2,4-DNP or 2,4-DNP-C3, could be caused by slower rates of these esters’ release from the vesicles and, as a result, their higher stability and biocompatibility.

Based on the literature [58,59], the lipid membrane in the gel phase is less permeable than in liquid crystalline. We were surprised to discover the fact that the change in the base bilayer lipid from eggPC to DSPC (increase in the phase transition temperature of the lipid membrane) did not reduce the rate of protonophore release from the liposomes in the case of esters with caproic acid. However, it worked for esters with palmitic acid. It can thus be suggested that, despite the relatively high entrapment efficacy, the 2,4-DNP esters with carboxylic acid of a chain length less than or equal to six are not optimal for the production of a long-circulated liposomal system with a significant sustained-release effect. These observations are consistent with the literature [32]. Thus, molecules with 1.7 < LogP_oct_ < 4 distribute between the lipid and aqueous phases and can easily release from the liposomes. Molecules with LogP_oct_ > 5 are entrapped almost completely in the lipid bilayer, and the release is significantly slower. Altogether, to design a stable liposomal formulation of 2,4-DNP ester with sustained-release properties, the optimal chain length of the ester should be close to the palmitic acid, and the lipid membrane could be varied and composed of phospholipids with a certain phase transition point depending on the desired release rate.

## 5. Conclusions

We have devised a procedure of synthesis of 2,4-DNP esters with carboxylic acids. This approach is quite universal and results in a high yield of esters with acids of various chain lengths. We demonstrated the dependence of the in vitro activity of the synthesized esters as mitochondrial uncouplers on the acid’s chain length. In the case of 4T1-Luc cells, the most pronounced effect of ATP synthesis inhibition was observed for the ester with a short-length chain acid. Moreover, this study has demonstrated the effective increase in the entrapment of 2,4-DNP into liposomes when esters were used. Here, we examined the dependence of the sustained release of 2,4-DNP from liposomes on the lipid composition and LogPoct of the ester. We showed that the optimal chain length of an ester should be close to the palmitic acid and the lipid membrane should be composed of phospholipids with a certain phase transition point depending on the desired release rate. Moreover, the increased effectiveness of the liposomal forms of esters compared to the free molecules in murine breast cancer 4T1-Luc cells and hepatocytes isolated from the normal liver AML12 cells was demonstrated. One of the significant factors of the final formulation efficacy is the liposome and ester stability.

Altogether, this work demonstrates a promising possibility for the liver-targeted delivery of the 2,4-DNP loaded into liposomes and provides a rationale for the design of 2,4-DNP esters and such liposomes for future biomedical applications as the treatment of non-alcoholic fatty liver disease.

## Data Availability

Not applicable.

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
