# Peer review of "Liposomal Form of 2,4-Dinitrophenol Lipophilic Derivatives as a Promising Therapeutic Agent for ATP Synthesis Inhibition"

_nanomaterials, 2022, doi:10.3390/nano12132162_

Round 1
Reviewer 1 Report
The paper aims to deliver the antidiabetic drug 2,4-DNP through liposomes in order to reduce its serious side effects. Liposomes are intended systemically administered ("for invasive administration" line 64) due to their tendency to achieve sustained release and accumulate in the liver.
The secondary aim of the work is to synthesize less active and therefore less toxic 2.4 DNP ester derivatives.
The idea behind the work is interesting even if the drug in question is very toxic and unpromising as an antidiabetic because of its toxicity. Parenteral administration via liposomes as an alternative to oral administration does not seem particularly attractive.
However, the work is well written and the experiments well described. There are few aspects to improve, listed below:
- The introduction should be rewritten because it is not faithful to the work: the dual objective of the work is not clearly described and it is not explicated why prodrugs of 2.4 DNP are prepared
- In determining the LogP it is necessary to use octanol saturated with water and water saturated with octanol. The solvent saturation step has not been described.
- Line 185: Authors should better describe the separation technique with the NAP-25 column which is not mentioned in the materials (what is it made of? Where was it purchased?)
- Line 198 replace "freed" with "free"
- Line 364-370 and fig 2. The luciferin-luciferase method is not well explained. “According to the method, we carried out experiment using 4T1-Luc cells expressing luciferase themselves. 2,4-DNP as a protonophore eliminates the mitochondrial proton gradient and inhibits the phosphorylation resulting in the reduce of ATP production [13]. addition due to decrease of ATP production. The kinetic curves profiles of the tested molecules are presented in Figure 2A ". In figure 2 compared to the control, an increase in the "relative luminescence units" can be observed
Author Response
- The introduction should be rewritten because it is not faithful to the work: the dual objective of the work is not clearly described and it is not explicated why prodrugs of 2.4 DNP are prepared
We thank the Reviewer for this comment. We agree with the comment and added the explanation why we chose liposomes and 2,4-DNP esters to the Introduction (Main document, Page 2 Line 74):
Such derivatives are prodrugs, as well as DNPME, with low cytotoxicity and do not show uncoupler activity, since the hydroxyl group is converted into an ester, which is not capable of transferring protons. Moreover, the gradual biodegradation of ester derivatives by esterases [33] leads to the prolonged release of 2,4-DNP in cells and, as a result, to the prolonged decrease of ATP synthesis. Steric structure and composition of the derivatives prone esters to incorporate into lipid membrane [34]. Since the more hydrophobic molecules interfere with the bilayer more strongly, they are also slowly release from liposomes, that provides the molecules stability and delivery to the target.
- In determining the LogP it is necessary to use octanol saturated with water and water saturated with octanol. The solvent saturation step has not been described.
We appreciate the Reviewer’s question. We added the solvent saturation step to the methods (Main document Page 5, line 169):
Before the experiment we mixed octanol and water in a flask, shook it vigorously and then waited for the two phases separation. After reaching of equilibrium the two phases are mutually saturated with one another.
- Line 185: Authors should better describe the separation technique with the NAP-25 column which is not mentioned in the materials (what is it made of? Where was it purchased?)
Thank you for this comment. We added description of the column to the methods (Page 5, line 196):
illustraTM NAP-25 column (Sephadex G-25 DNA Grade, GE Healthcare, UK).
- Line 198 replace "freed" with "free"
We thank the Reviewer for this comment and made relevant correction.
- Line 364-370 and fig 2. The luciferin-luciferase method is not well explained. “According to the method, we carried out experiment using 4T1-Luc cells expressing luciferase themselves. 2,4-DNP as a protonophore eliminates the mitochondrial proton gradient and inhibits the phosphorylation resulting in the reduce of ATP production [13]. addition due to decrease of ATP production. The kinetic curves profiles of the tested molecules are presented in Figure 2A ". In figure 2 compared to the control, an increase in the "relative luminescence units" can be observed
We appreciate the Reviewer’s comment. We studied the kinetics of luciferin-luciferase reaction in the details on Pages 10-13. Also, we added the suggested explanation in the Results (Page10, Line 374):
According to the method, we followed luminescence signal produced during the reaction of luciferase expressed by 4T1-Luc cells themselves and D-luciferin as a substrate. The enzymatic reaction depends on the ATP content.
Reviewer 2 Report
This work developed a liposomal form of 2,4-dinitrophenol lipophilic derivatives to provide liver-targeted delivery and sustain release 2,4-dinitrophenol. The work provided rationale for the design of 2,4-DNP esters and such liposomes for future biomedical applications as the treatment of non-alcoholic fatty liver disease. The following point should be addressed.
1. Make sure the line numbers are arranged correctly. (For example, line 142-148 were arranged in the table)
2. The unit and Latin terms need to be checked and corrected. (For example, mL, 1×104 cells/well, in vitro…)
3. In luciferin-luciferase coupled assay, choosing 4T1-Luc as testing cell line may be reasonable. But is it a good choice for cytotoxicity test? 4T1 is Murine breast cancer cell, could it represent in the study (non-alcoholic fatty liver disease)?
4. Figure 1. Cytotoxicity of compounds on (A) 4T1-Luc cells and on (B) AML12:
Though you mentioned the IC50 value is similar to the literature estimated IC50 value, are the standards deviation acceptable (especially 2,4-DNP-C6 in AML12 at 50 µM) and won’t affect the accuracy of interpretation?
5. In page 15, it was mentioned that “According to the cytotoxicity test (see 480 Supplementary S-Figure 18A and 18D), encapsulation of compound 1 and compound 2 led to the increase in toxicity, probably, due to the increase in bioavailability of the tested molecules.” Could you please explain in more detail about why it is not because of the cytotoxicity of compounds themselves are higher?
Author Response
- Make sure the line numbers are arranged correctly. (For example, line 142-148 were arranged in the table)
We greatly appreciate the Reviewer’s suggested improvements and made relevant corrections.
- The unit and Latin terms need to be checked and corrected. (For example, mL, 1×104cells/well,in vitro…)
We greatly appreciate the Reviewer’s suggested improvements and made relevant corrections.
- In luciferin-luciferase coupled assay, choosing 4T1-Luc as testing cell line may be reasonable. But is it a good choice for cytotoxicity test? 4T1 is Murine breast cancer cell, could it represent in the study (non-alcoholic fatty liver disease)?
We appreciate the reviewer’s question. During our study we tested two cell lines: 4T1-Luc, which we used for pre-screening and more detailed studies, and AML12 cells of hepatocytes isolated from the normal liver to check the effectiveness of studied molecules and formulations in liver diseases treatment.
- Figure 1. Cytotoxicity of compounds on (A) 4T1-Luc cells and on (B) AML12:
Though you mentioned the IC50 value is similar to the literature estimated IC50 value, are the standards deviation acceptable (especially 2,4-DNP-C6 in AML12 at 50 µM) and won’t affect the accuracy of interpretation?
We thank the Reviewer for this comment. At Figure 1 we combined the concentration-response curves of 3-5 independent experiments and calculated the StD. IC50 values are estimated from the individual experiments and the average±StD for each molecule is performed:
for 4Т1 cells line IC50 were following: 2,4-DNP 352±8 mM, 2,4-DNP-C3 122±35 mM, 2,4-DNP-C6 166±21 mM, 2,4-DNP-C16 218±77 mM.
for АML12 cells line IC50 were following: 2,4-DNP 875±333 mM, 2,4-DNP-C3 212±84 mM, 2,4-DNP-C6 214±41 mM, 2,4-DNP-C16 224±42 mM.
In the case of AML12 the StD is high due to we did not reach lower than 40% of cells viability that impedes the approximation and estimation.
- In page 15, it was mentioned that “According to the cytotoxicity test (see 480 Supplementary S-Figure 18A and 18D), encapsulation of compound 1 and compound 2 led to the increase in toxicity, probably, due to the increase in bioavailability of the tested molecules.” Could you please explain in more detail about why it is not because of the cytotoxicity of compounds themselves are higher?
We appreciate the Reviewer’s comment. We compared cytotoxicity of free molecules and their liposomal forms at the same concentrations. Tests resulted in higher cytotoxicity of liposomal form. Free liposomes possess a low cytotoxicity [doi: 10.2147/IJN.S207589]. Also, liposomes loaded with a compound, increase an entrapment of molecule by cells. This effect means that at the same concentrations of added free ester and its liposomal form to cells, in the case of liposomes more active molecule will be entrapped by cells for a certain time point (in our case, 24 h) that results in higher cytotoxicity.